



# The significance of vertical moisture diffusion on drifting Snow sublimation near snow surface

Ning Huang[1] and Guanglei Shi[1]

1Key Laboratory of Mechanics on Disaster and Environment in Western China, Lanzhou University,

Lanzhou 730000, China

*Correspondence to*: Guanglei Shi (shigl14@lzu.edu.cn)

**Abstract.** Drifting snow sublimation is a physical process containing phase change and heat change of the drifting snow, which is not only an important parameter for the studying of polar ice sheets and glaciers, but a significant one for the ecology of arid and semi-arid lands, where snow cover is the main fresh water resource. However, in the previous studies drifting snow sublimation near surface was ignored. Herein, we built a drifting snow sublimation model containing vertical moisture diffusion equation and heat balance equation, to study drifting snow sublimation near surface. The results showed that though drifting snow sublimation near surface was strongly reduced by negative feedback effect, relative humidity near surface didn't reach the saturation state caused by vertical moisture diffusion. Therefore, the sublimation near surface will not stop in drifting snow near surface. The sublimation rate near surface is 3-4 orders of magnitude higher than that at 10 m. And the mass of snow sublimation near surface accounts for even more than half of the total if the wind velocity is small. Therefore, drifting snow sublimation near surface can't be neglected.

## 1 Introduction

The polar ice sheets, mountain glaciers, snowy area in high latitude of Northern Hemisphere

(such as North of Canada, Greenland, etc), whose main source is snow, have profound influence on

the global hydrologic cycle, climate change and ecological system. Extensive researches showed that

drifting snow sublimation was an important method to change the snow distribution, especially in the

polar ice sheets, highland mountains and high latitude of Northern Hemisphere. For example,

Pomeroy and Jone (1995) found that the mass of drifting snow sublimation was equal to 18.3% of

annual precipitation in coastal Antarctica; while Liston and Sturm (2004) found that it was equal to 22%

of winter precipitation in Arctic Alaska. Pomeroy and Essery (1999) found that blowing snow

sublimation fluxes during blowing snow return 10±50% of seasonal snowfall to the atmosphere in

North American prairie and arctic environments. MacDonald et al. (2010) found that the mass of

drifting snow sublimation was equal to 17%-19% of annual precipitation in Rocky Mountains,

Canada. Zhou et al. (2014) pointed out that the mass of drifting snow sublimation was equal to 24%



of annual precipitation in western Chinese mountains. These results indicate that drifting snow
sublimation is very important to the study of global and polar hydrological systems.
Some scientists directly measured drifting snow sublimation using eddy covariance, but this
method can only obtain a few points of information, and it is difficult to predict the whole sublimation
in snowy areas (Pomeroy and Essery, 1999; Cullen et al., 2007; Marks et al., 2008; Reba et al., 2012).
Therefore, there is a high demand of studying the sublimation of snow using numerical model.
The sublimation of snow particles in the drifting snow is normally accompanied with heat
absorption and water vapor production, which will cause a decrease in the ambient air temperature
and an increase in humidity. The increased humidity will in turn inhibit the sublimation of snow
particles; while the lower temperature will lead to a decrease in the air saturated vapor pressure,
which will also inhibit the snow sublimation. Many researchers believed that the sublimation of snow
particles near surface would occur violently at the early stage of drifting snow process, since the high
concentration of snow particles near surface would result in a rapid air temperature decrease and
humidity increase. Then the humidity would reach saturation quickly near surface, and the
sublimation would stop at the saturated layer of humidity. Therefore, the snow sublimation near
surface was negligible in the fully developed drifting snow (Déry et al., 1998; Bintanja, 2001a; Mann
et al., 2000). However, some researchers found that humidity near surface didn't reach saturation in
the drifting snow in the field or wind tunnel experiments, which they thought was caused by water
transport (convection and diffusion) (Schmidt, 1982; Groot Zwaadtink et al., 2011). Déry and Yau
(1999) fix the relative humidity at 95% instead of 100% at the surface when they simulated the
blowing snow sublimation. They found that the time-integrated values of sublimation increased 14%
than the results which fix the relative humidity at 100%, so humidity near surface is very important
for the simulations of blowing sublimation. Huang et al. (2016) calculated the snow sublimation in
the saltation layer, taking into consideration of the effect of horizontal moisture convection on the
non-homogeneous snow cover. Their results showed that drifting snow sublimation in the saltation
layer could not be neglected in the presence of horizontal moisture convection. But they did not
discuss the sublimation near surface of areas such as polar ice sheets, grassland covered by snow, etc.,
where the snow cover was very large and the water convection was very weak. Therefore, studies on
the snow-sublimation in these regions are of great significance for the understanding of global
hydrological systems and ecosystems.



However, in the previous blowing snow sublimation model, the diffusion equation was often
used to describe the movement of snow particles, which can describe the movement of small particles
well. But the diffusion equation is difficult to describe the movement of large snow particles which
are mainly distributed in the near surface area (Déry et al., 1998; Xiao et al., 2000; Vionnet et al.
2014). Huang et al. (2016) used the Lagrangian particle tracing method to describe the movement of
near-surface snow particles, and for the first time calculated the sublimation of saltation particles in
near surface region on non-uniform snow cover. But this model can not describe snow particles
suspending in upper air. Furthermore, all above exiting models did not take into consideration of the
effects of vertical moisture diffusion on the sublimation.
Therefore, a drifting snow model has firstly been built to describe the movement of snow
particles of both saltating near surface and suspending in the higher region. Then, a drifting snow
sublimation model has been built the combination of the drifting snow model, a vertical moisture
diffusion equation and a heat balance equation. Then drifting snow sublimation with three wind
speeds was calculated. The temporal evolution and vertical profiles of temperature, relative humidity,
mass concentration of snow particles, snow sublimation rate were analyzed in details. Meanwhile, the
proportions of the sublimation mass of saltation snow grains and saltation layer to the total
sublimation mass were also given.
**2    Method**
**2.1 Basic Equations of the Flows**
The horizontal wind field satisfies the Navier–Stokes equation at the atmospheric boundary layer.
Considering a fully developed steady flow field on an infinite polar ice sheet where the changes of
wind field in the lateral and flow direction are negligible, the fully developed horizontal direction
flow field equation can be obtained according to the theory of mixing length by Prandtl.
$$\frac{\partial}{\partial z}(\rho_a \kappa^2 z^2 \left|\frac{du}{dz}\right|\frac{du}{dz}) + F = 0 \qquad (1)$$
where $\kappa$ is the von Karman constant, $\rho_a$ is air density, u is the horizontal wind speed and F is the
reaction force of the snow particle on the flow field.





### 2.2 Snow particle motion equation

The snow particles jumping from the bed are divided into saltation and suspended particles when calculating snow particle movement. These two types of particles are distinguished based on the particle size and flow field conditions. Then the saltation particles are calculated by Lagrange particle tracing method, and the suspension particles are calculated by diffusion equation.

#### 2.2.1 Judging criteria of saltation and suspended particles

The judging criterion of saltation and suspended particles is as follows (Scott, 1995):

$$
\begin{cases}
w_s/(ku_*) > 1, & \text{saltation particle} \\
w_s/(ku_*) \leq 1, & \text{suspension particle}
\end{cases}
\tag{2}
$$

where $u_*$ is the friction velocity and $w_s$ is the final sedimentation velocity of the particles (Carrier, 1953):

$$
w_s = -\frac{A}{D} + \sqrt{\left(\frac{A}{D}\right)^2 + BD}
$$
$$
A = 6.203 \upsilon_a
\tag{3}
$$
$$
B = \frac{5.516 \rho_p}{8 \rho_a} g
$$

where D is diameter of snow particle, $\upsilon_a$ is air viscosity coefficient, $\rho_p$ is the densities of snow particle, $g$ is the acceleration of gravity.

#### 2.2.2 Basic equations of saltation particles

Saltation particle motion equation is as follows (Huang et al., 2011):

$$
m\frac{dU_p}{dt} = F_D\left(\frac{U_a - U_p}{V_r}\right)
\tag{4}
$$

$$
m\frac{dV_p}{dt} = -G + F_B + F_D\left(\frac{V_a - V_p}{V_r}\right)
\tag{5}
$$

$$
\frac{dx_p}{dt} = U_p
\tag{6}
$$



$$\frac{dy_p}{dt} = V_p \qquad (7)$$

where $m$ is the mass of snow particle, $G$ is the gravity of snow particle, $U_a$ and $V_a$ are the
horizontal and vertical velocity of air, $U_p$ and $V_p$ are the horizontal and vertical velocities of snow
particle, $V_r = \sqrt{(U_p - U_a)^2 + (V_p - V_a)^2}$ is the relative velocity of movement of the snow particles
and the flow field, $F_B$ and $F_D$ are the buoyancy and traction forces of snow particles, $x_p$ and $y_p$
are the horizontal and vertical positions of snow particles.
The splash function fitted by Sugiura and Maeno (2000) according to the observations of the low
temperature wind tunnel experiment was chosen,
$$S_v(e_v) = \frac{1}{b^a G(a)} e_v^{a-1} \exp\left(-\frac{e_v}{b}\right) \qquad (8)$$

$$S_h(e_h) = \frac{1}{\sqrt{2\pi\sigma^2}} \exp\left[-\frac{(e_h - \mu)^2}{2\sigma^2}\right] \qquad (9)$$

$$S_e(n_e) = {}_m C_{n_e} p^{n_e} (1-p)^{m-n_e} \qquad (10)$$

where $S_v(e_v)$, $S_h(e_h)$ and $S_e(n_e)$ are the probability distribution functions of the vertical
restitution coefficient $e_v$, horizontal restitution coefficient $e_h$, and the number of grains ejected $n_e$.
**2.2.3 Basic Equations of Suspended particles**
The movement of suspension particles is described by the following vertical diffusion equation
according to horizontal uniformity condition,
$$\frac{\partial q}{\partial t} = \frac{\partial}{\partial y}\left(K_s \frac{\partial q}{\partial y} + w_s q\right) + S \qquad (11)$$

where q is the snow particle mass concentration, $K_s$ is the vertical diffusion coefficient, S is the
volume sublimation rate of snow grain. $K_s = \delta \kappa u_* z$, $\delta$ is as follows (Csanady, 1963),
$$\delta = \frac{1}{\sqrt{1 + \frac{\beta^2 f^2}{w_a^2}}} \qquad (12)$$

where $\beta$ is the proportionality constant, $w'$ is the turbulent fluid velocity in the vertical, and we set





$\beta = 1$,   $\overline{w'^2} = u_*^2$ ..

### 128     2.2.4 Aerodynamic Entrainment

The aerodynamic entrainment equation of Shao and Li (1999) is chosen,
$$N_a = V u_* \left( 1 - \frac{u_{*_t}^2}{u_*^2} \right) D^{-3} \tag{13}$$

where $N_a$ is the number of snow particles taking off causing by aerodynamic entrainment, $\varsigma$ is a
non-dimensional coefficient, approximately equal to $1 \times 10^{-3}$, $u_*$ is the friction velocity, $u_{*_t}$ is
the threshold friction velocity.

### 134     2.3 Sublimation formula

The sublimation formula is as follows (Thorpe and Mason, 1966),
$$\frac{dm}{dt} = \frac{\pi D (RH - 1)}{\frac{L_s}{K N u T_a} \left( \frac{L_s}{R_v T_a} - 1 \right) + \frac{R_v T_a}{Sh K_l e_s}} \tag{14}$$

where $RH$ is the relative humidity of air, $T_a$ is air temperature, $L_s$ is the latent heat of sublimation
(equal to $2.84 \times 10^6$ J kg$^{-1}$), $K_a$ is the thermal conductivity of air, $R_v$ is the gas constant of water
vapor (equal to 461.5 J kg$^{-1}$ K$^{-1}$), $K_l$ is the molecular diffusion of water vapor of atmosphere, $e_s$ is
the saturated vapor pressure relative to the ice surface. $Nu$ and $Sh$ are the Nusselt and Sherwood
numbers (Thorpe and Mason, 1966; Lee, 1975),
$$Nu = Sh = \begin{cases} 1.79 + 0.606 \, \text{Re}^{0.5} & 0.7 < \text{Re} \leq 10 \\ 1.88 + 0.580 \, \text{Re}^{0.5} & 10 < \text{Re} < 200 \end{cases} \tag{15}$$

where $R_e = \frac{D V_r}{\upsilon_a}$ is Reynolds number.

### 144     2.4 Heat and humidity equation

The heat and humidity equations of air are as follows (Déry and Yau, 1999; Bintanja,2000),
$$\frac{\partial \theta}{\partial t} = \frac{\partial}{\partial z} \left( K_\theta \frac{\partial \theta}{\partial z} \right) - \frac{L_s S}{\rho_f C} \tag{16}$$

$$K_\theta = \kappa u_* z + K_T \tag{17}$$



$$\frac{\partial h_u}{\partial t} = \frac{\partial}{\partial z}\left(K_q \frac{\partial h_u}{\partial z}\right) + \frac{S}{\rho_f} \qquad (18)$$

$$K_h = \kappa u_* z + K_V \qquad (19)$$

where $K_T$ and $K_V$ are the molecular diffusion coefficients of heat and water vapor, C is the specific
heat of air.

**2.5 Initial and boundary conditions**

The initial potential temperature $\theta_0 = 263.15K$, and the initial absolute temperature is
$$T_0 = \theta_0 \left(\frac{p}{p_0}\right)^{0.286} \qquad (20)$$

Where p is atmospheric pressure, its initial value is
$$p = p_0 \exp\left(-\frac{yg}{R_d \theta_0}\right) \qquad (21)$$

where $p_0 = 1000hpa$, $R_d = 287 JKg^{-1}K^{-1}$ is the gas constant for dry air.
The initial relative humidity profile is
$$RH = 1 - R_s \ln(z / z_0) \qquad (22)$$

where $z_0$ is the surface roughness, and its value is $3 \times 10^{-5} m$ at snow bed (Nemoto and Nishimura,
2001), and $R_s = 1.9974 \times 10^{-2}$.
The conversion relationship of relative humidity and specific humidity is
$$q = 0.622 \cdot \frac{e_s}{p - e_s} \cdot RH \qquad (23)$$

where $e_s = 610.78 \exp\left[21.87\left(T - 273.16\right)/\left(T - 7.66\right)\right]$.
The calculation area is set to 1 m in length, 10 m in height, and 0.01 m in width. The time step is
$10^{-5}$ s for saltation particles, $10^{-2}$ s for suspended particles, $10^{-3}$ s for wind, and the calculation time is
1500 s. The motion of saltation particles is only calculated for 10 s in consideration of the practical
simplicity, since saltation particles will stabilize within a few seconds. The data of saltation particles
in the air and the jumping particles from bed are then replaced by the data averaged in 10 s. The
threshold friction velocity is 0.21 m/s (Nemoto and Nishimura, 2001).
The snow particle size distribution fits the results of Schmidt (1982) field observations (Fig. 1).



3    **Results and Discussion**

In order to verify the judging criteria in eq.2, we divided the particles into sets varied by 10 $\mu m$

(1-600 $\mu m$), and used eq.16 to simulate all the jumping particles. Then we accumulated the mass of
snow particles in the air from small to large particles until the mass was equal to 99.9% of the total
mass of snow particles in the air, and the particle diameter $D_{99\%}$ was recorded. $D_{99\%}$ and threshold
particle diameter $D_{th}$ calculated by eq.2 were compared, and the results is shown in Table1.

As shown in Table 1, particles which are larger than the threshold particle do not enter into air

according to the vertical diffusion, indicating that these particles can not be described by the diffusion
equation. Thus, the judging criteria in eq.2 are reliable.

In order to verify the reliability of the blowing snow model in this paper, we compared our mass

concentration results with that of the field observations (Fig.2). The red dot in Fig. 2 is the field
observation results near Saskatoon, Canada in 26 January 1987 (Pomeroy and Male, 1992). And the
black line in Fig.2 is our numerical simulation results using the same conditions with the above filed
observation results. It is shown that our simulation results are basically consistent with those observed
in the field, which demonstrates the reliability of our simulations.

We also compared our sublimation results with that of the field observations to verify their

reliability (Fig.3). The red lines in Fig. 3 are the results gotten from the observed data by Schmidt
(1982) in Wyoming, U.S.A, in 1982. The black line was the simulated results using the same
environmental conditions as those of Schmidt's. It can be seen that the total sublimation rates
calculated by the model of this paper (black line) are approximately the same as Schmidt's results,
and the sublimation rate at 0.01 m was two orders of magnitude larger than that at 0.1 m. These
results demonstrate that our snow sublimation results are reliable too.

We further compared our results with corresponding results of other models under the same

conditions. The black line in Fig. 4 is the result of the suspension particles sublimation rate calculated
by our model ($u_* = 0.89, T = 253.15K$). The other four lines are the results calculated by Xiao et al.
(2001) using four existing blowing snow sublimation models, in which the sublimation of saltation
particles near surface was neglected. It is shown from Fig. 4 that all the rates of suspension particle
increase with height first, and then start to decrease, and the peak is at about 0.1 m. The results of this
paper are higher than that of Xiao et al. (2001). The peaks of total sublimation rate using our model



and Schmidt (1982) are all at a height about 0.01 m, which is lower than that of the four blowing
snow models in Fig. 4. But the values of peak in this paper and Schmidt (1982) are two orders of
magnitude larger than that of the four blowing snow models. This is because the sublimation of
saltation particles is neglected in the four models, which is the main movement of snow particles near
surface.
Fig. 5 is the temporal evolution of the mass of saltation particles and suspended particles versus
various friction velocities. It is shown that the mass of saltation and suspended particles increase with
time, and finally reach steady. The mass of saltation particles is much larger than that of suspension
particles in the steady state. The time for saltation particles to reach steady state is about 2 s, and
about 300 s for suspended particles.
Fig. 6 shows the curves of temperature and humidity with height in the near-surface region of
saltation particles and they are compared with their initial conditions. It is shown that drifting snow
sublimation changes air temperature and relative humidity, and the change amplitude increases with
the friction velocity. This is because the larger the friction velocity is, the more snow particles in the
air are, and the more sublimation will occur, which makes a greater impact on temperature and
humidity.
We compared the temperature and humidity with height. It is shown in Fig. 7 and 8 that the
change amplitude of temperature and relative humidity increases while the height decreases.
Combined with the results from Fig. 10, the mass concentration of snow particles increases while
height decreases, which can make a stronger sublimation.
It is shown in Fig. 8 that the time for humidity to reach steady is about 2 s at 0.01 m, which is
consistent with the stability time of saltation snow particles; and at 10 m is about 300 s, which is
consistent with the stability time of suspension snow particles. This is because the main part of snow
particles near surface is saltation particles, opposite to that in upper air which is mainly suspension
particles (Fig. 10).
Fig. 8 shows that the relative humidity near surface with three kinds of friction velocities does
not reach saturation when the blowing snow reaches steady, which indicates that the snow sublimation
does not stop. It also shows that the vertical diffusion of water vapor can reduce the negative feedback
effect effectively.





It can be seen from Fig. 9a that the sublimation rate of saltation particles increases with time first,
then starts to decrease, in which the peak is at about 2 s and finally reaches stability at about 300 s.
The negative feedback effect on saltation particles is very obvious and the time to reach a steady state
is about 300 s. Because the mass of saltation particles increases with time during the first 2 s, and the
increasing amplitude of which is larger than that of relative humidity, and the saltation sublimation
rate increases with time. However, the mass of saltation particles basically stay unchanged after 2 s,
while the relative humidity near surface gradually increases. Therefore, the sublimation rate decreases
with time. The relative humidity near surface also reaches steady after 300 s, which results in the
stability of sublimation rate. The saltation particles distribute mainly near surface, where the change
amplitude of relative humidity is strong which results in a strong negative feedback effect on saltation
particles.
It is shown in Fig. 9b that sublimation rate of suspended particles increases with time and
finally reaches steady at about 300 s. The negative feedback effect on suspended particles is not
obvious. The mass of suspension particles increases with time during the first 300 s, which the
increase amplitude of is larger than that of relative humidity, so the suspension sublimation rate
increases with time. Then the mass of suspended particles and relative humidity both reach stable,
which leads to the sublimation rate of suspended particles reaching stable. Since the suspended
particles mainly distribute in upper air where the change amplitude of relative humidity is weak, the
negative feedback effect on suspended particles is not strong.
Although the effect of negative feedback on saltation particles is stronger than suspended
particles, the sublimation rate of saltation particles is still greater than that of suspended particles,
indicating that the sublimation of saltation particles is very strong even under the effect of negative
feedback.
Fig. 10 shows that the mass concentration of snow particles increases with friction velocity and
decreases with height, and the mass concentration of saltation particles is much higher than that of
suspended particles. It can be seen from Fig. 10a that saltation particles mainly distribute below 0.1 m,
which is consistent with the previous experimental results (Takeuchi, 1980).
Fig. 11 shows that sublimation rates increases with friction velocity. Sublimation rates of
saltation and suspended particles increase with height first, then start to decrease. The peak of



saltation particles is at about 0.01 m, and that of suspended particles is at about 0.1 m. This is because
the mass concentration and relative humidity of snow decrease with height, while temperature
increases. However, mass concentration of saltation particles changes more strongly than that of
suspension particles with height. Therefore, sublimation rate of saltation particles reaches peak at
lower height.
Table 2 shows that the sublimation rate at 0.01 m is two orders of magnitude faster than that at
0.1 m, which is same as the experimental results in Fig. 3, and it's 3-4 times faster than that at 10 m,
although the negative feedback effect near surface is stronger than other regions. Because the mass
concentration of snow particles near surface is much higher than that in other regions (Fig. 8), and
water vapor near surface is not saturated, the sublimation rate near surface is much faster than that in
other regions.
In the previous studies the snow sublimation near surface was ignored. That is, to define a wind
velocity related height, below which saltation particles move. Then assumed that moisture in the
region was saturated and therefore the snow sublimation would not be counted (Déry et al., 1998;
Xiao et al. 2000; Vionnet et al. 2014). Three heights at several wind velocities proposed by Déry et al.
(1998), Pomeroy and Male (1992), and Xiao et al. (2000) were respectively given in Table 3 (The
height by Vionnet et al. was the same as that of Pomeroy and Male). Fig. 12 shows the actual ratio of
our simulated sublimation mass below the three heights to the total. It is shown that all the
sublimation masses below three heights account for more than half of the total sublimation mass. This
is because the main part of snow particles is saltation particles (Mellor, 1965), which mainly
distribute in near surface region. And although sublimation near surface leads to significant changes
of temperature and humidity, which have a strong inhibition effect on sublimation, moisture near
surface does not reach saturation due to the vertical diffusion of water vapor, which results in
continuous snow sublimation. Therefore, the main part of the mass of sublimation is sublimation of
saltation particles, and the previous methods neglecting blowing snow sublimation near surface is not
appropriate. Fig. 12 also shows that the proportion of the sublimation mass near surface decreases
with friction velocity. Because more snow particles can enter into upper air with increased wind
velocity, which will lead to decreasing proportion of snow particles near surface, the proportion of the
mass of sublimation near surface will decrease as well.



Fig.13 shows the vertical profiles of vapor flux. It is shown that vapor flux increases rapidly in near surface region, where most of saltation particles move, then slows down greatly after reaching a certain height. For there is no horizontal flux of water vapor,the water vapor flux at any height must be equal to the total amount of water vapor generated per second below the height. So most of the water vapor is coming from near surface regions. From Fig. 13 it can also be seen that vapor flux increases with friction velocity, for humidity (Fig.5) and moisture diffusion coefficient (eq.17) increase with friction velocity.

## 4  Conclusions

We have established a blowing snow sublimation model, which includes vertical moisture diffusion and heat balance, to study the snow sublimation near surface in large snow cover area in this paper. The simulation results showed that the blowing snow sublimation decreased air temperature and increased humidity of air. Meanwhile, the snow sublimation was reduced by the negative feedback effect of temperature and humidity, especially for near surface, which is in agreement of previous researches. However, moisture near surface was not saturated due to the vertical moisture diffusion, so snow sublimation near surface continued. The sublimation rate near surface was even larger than that in the upper air, because mass concentration of snow particles near surface was much higher than that in other regions. The sublimation rate at 0.01 m is two orders of magnitude greater than that at 0.1 m, and is 3-4 orders of magnitude greater than that at 10 m. Furthermore, when the wind speed was low, the mass of sublimation near surface accounted for more than half of total mass of sublimation, and could not be neglected. Most of the air vapor in bellowing snow is form near surface region. Therefore, blowing snow sublimation near surface should be taken seriously in the study of snow sublimation and water vapor transport in the future.

*Acknowledgements*. This work is supported by the State Key Program of National Natural Science Foundation of China (91325203), the National Key Research and Development Program of China (2016YFC0500900), and the Innovative Research Groups of the National Natural Science Foundation of China (11121202).

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





**Table 1: Comparison of $D_{th}$ and $D_{99\%}$**

|  | $u_* = 0.35ms^{-1}$ | $u_* = 0.41ms^{-1}$ | $u_* = 0.54ms^{-1}$ |
|---|---|---|---|
| $D_{th}$ | 80.55μm | 87.84μm | 102.61μm |
| $D_{99\%}$ | ≤80μm | ≤90μm | ≤110μm |

**Table 2: Sublimation rate at 1500s for various heights (*: friction velocity (m/s); **: height (m); ***: sublimation rate (kgm$^{-3}$s$^{-1}$))**

|  | $u_* = 0.35ms^{-1}$ | $u_* = 0.45ms^{-1}$ | $u_* = 0.55ms^{-1}$ |
|---|---|---|---|
| h=0.01[**] | 3.71E-04[***] | 4.05E-04 | 4.21E-04 |
| h=0.05 | 1.22E-05 | 2.31E-05 | 3.18E-05 |
| h=0.1 | 6.11E-07 | 3.08E-06 | 5.37E-06 |
| h=1 | 1.68E-07 | 1.12E-06 | 2.29E-06 |
| h=5 | 2.93E-08 | 2.88E-07 | 7.52E-07 |
| h=10 | 8.44E-09 | 1.09E-07 | 3.31E-07 |

**Table 3: Height which most of saltation particles distributed below for various friction velocities**

|  | $u_* = 0.35ms^{-1}$ | $u_* = 0.45ms^{-1}$ | $u_* = 0.55ms^{-1}$ |
|---|---|---|---|
| **Déry et al. (1998)** | 0.0196m | 0.0253m | 0.0316m |
| **Pomeroy and Male(1992)** | 0.0222m | 0.0306m | 0.0395m |
| **Xiao et al.(2000)** | 0.05m | 0.05m | 0.05m |



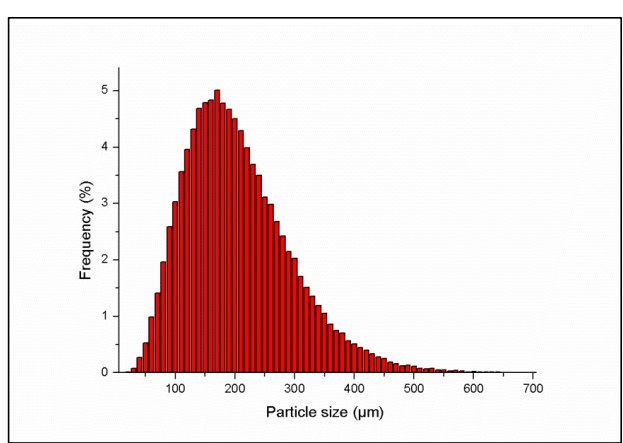

**Figure 1: Particle size distribution**





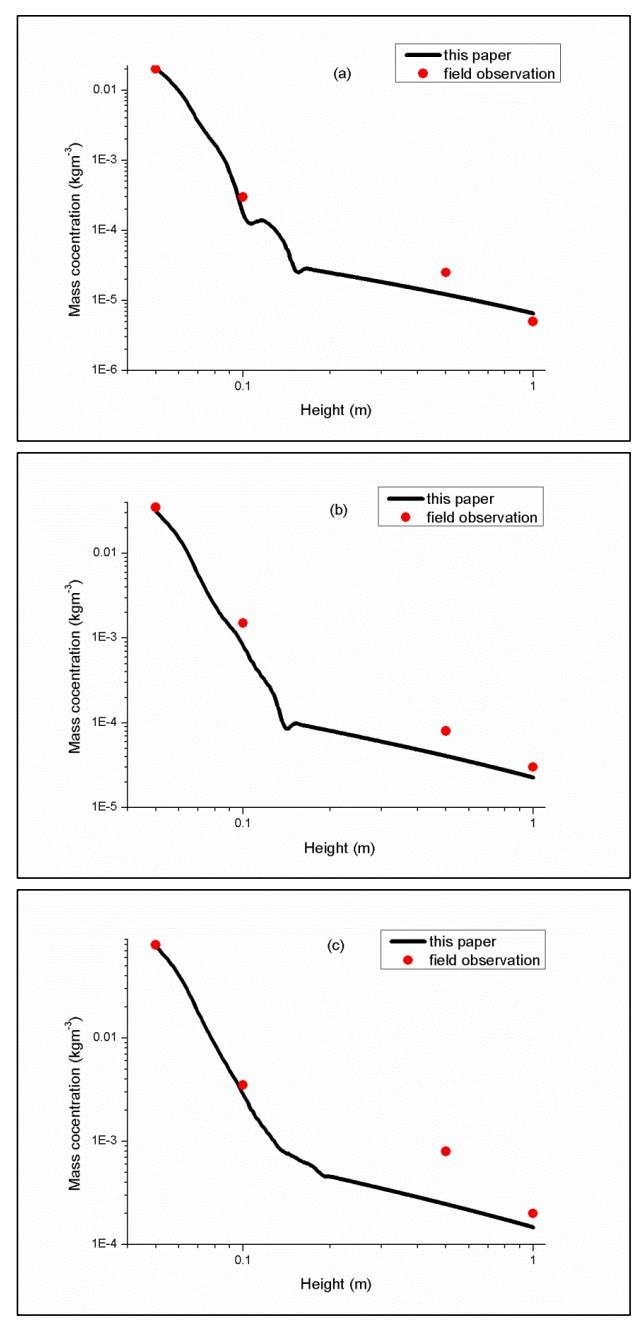

Figure 2: Comparison of mass concentration for this paper and field observation (a: $u_* = 0.35ms^{-1}$ ; b:

$u_* = 0.41ms^{-1}$ ; c: $u_* = 0.54ms^{-1}$ )



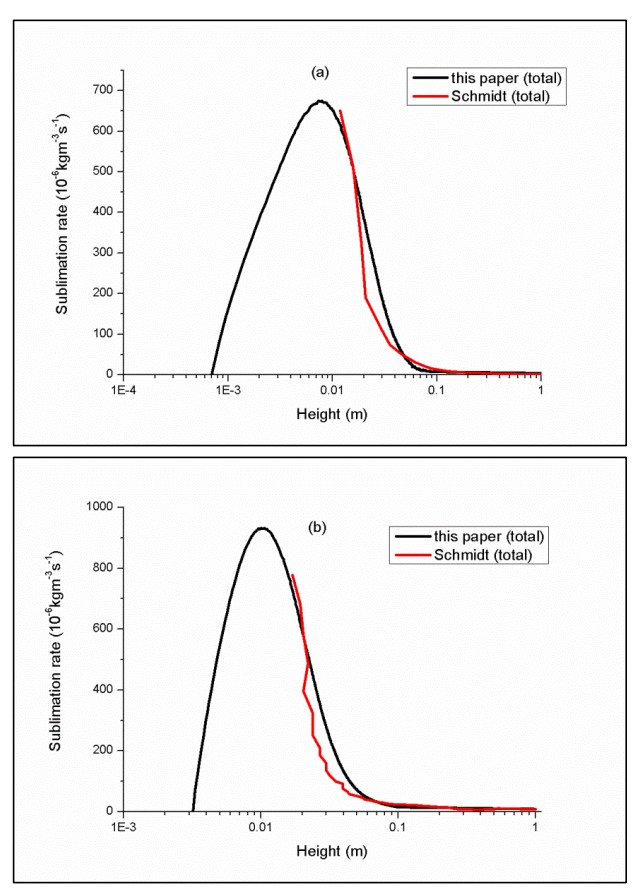

**Figure 3: Comparison of sublimation rate for this paper and Schmidt (1982) (a:** $u_* = 0.632ms^{-1}, T = 267.45k$ **;**

**b:** $u_* = 1.072ms^{-1}, T = 265.65K$ **)**





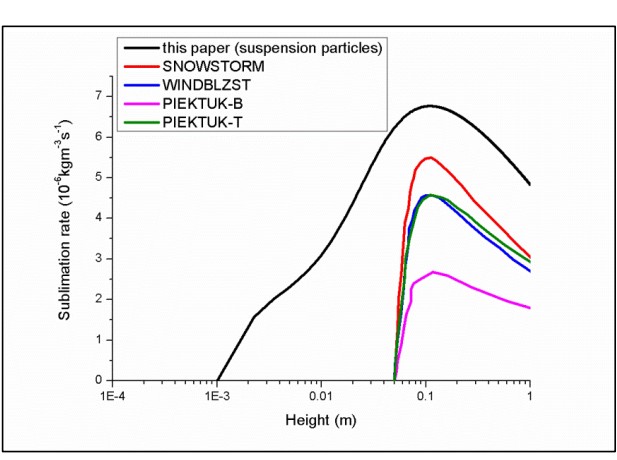

Figure 4: Comparison of sublimation rate for this paper and four blowing snow's models (Xiao et al., 2000)





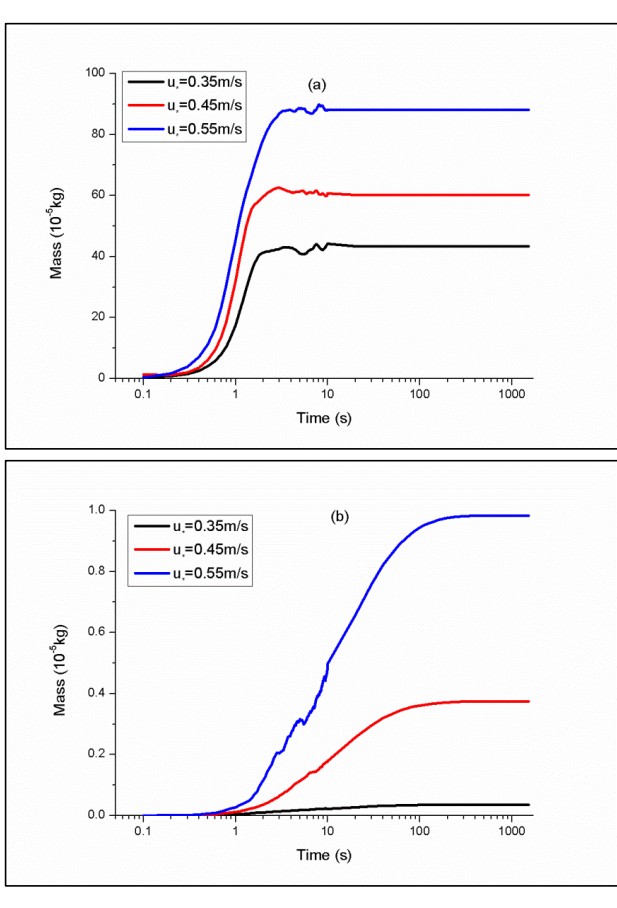

**Figure 5 : Temporal evolution of mass of saltation particles and suspension particles (a: saltation particles;**

**b: suspended particles)**





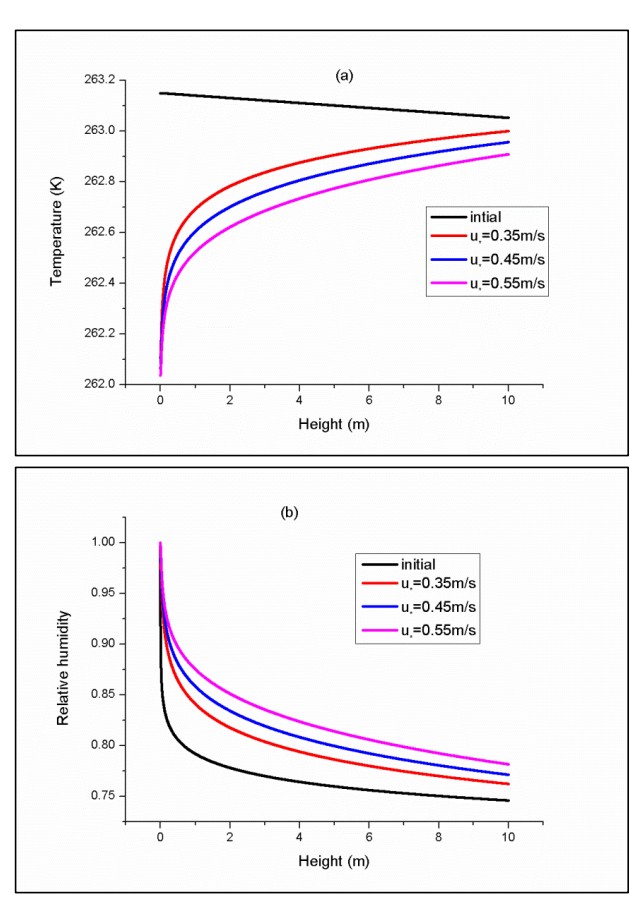

**Figure 6: Vertical profiles of temperature and relative humidity**

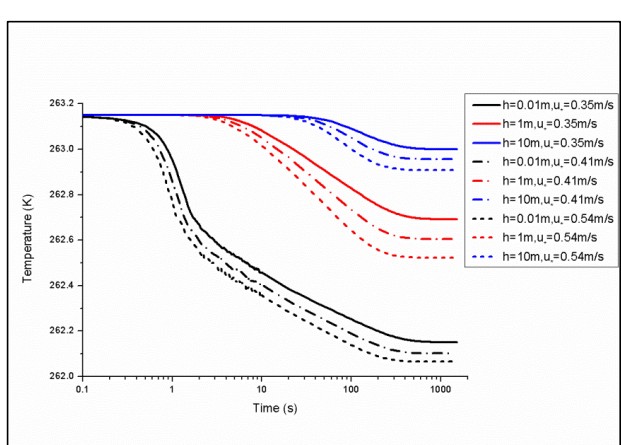

Figure 7: Temporal evolution of temperature for various heights





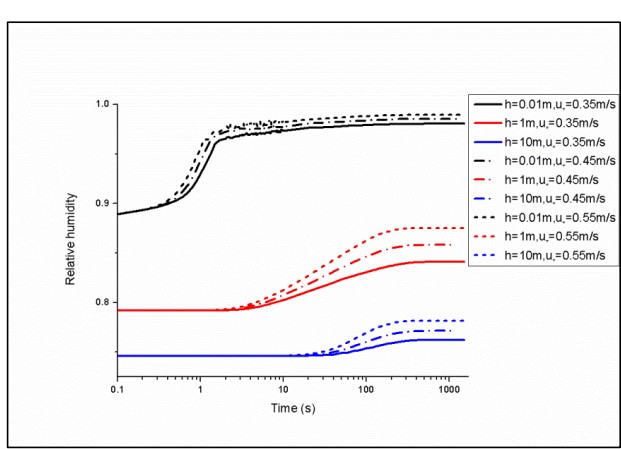

Figure 8: Temporal evolution of relative humidity for various heights



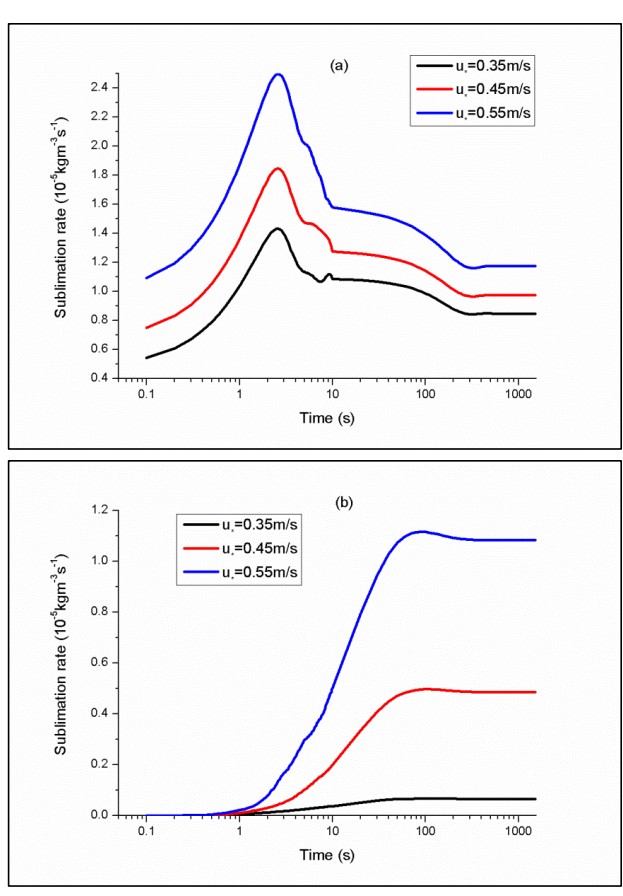

**Figure 9: Temporal evolution of saltation sublimation rate and suspension sublimation rate(a: saltation particles; b: suspended particles)**

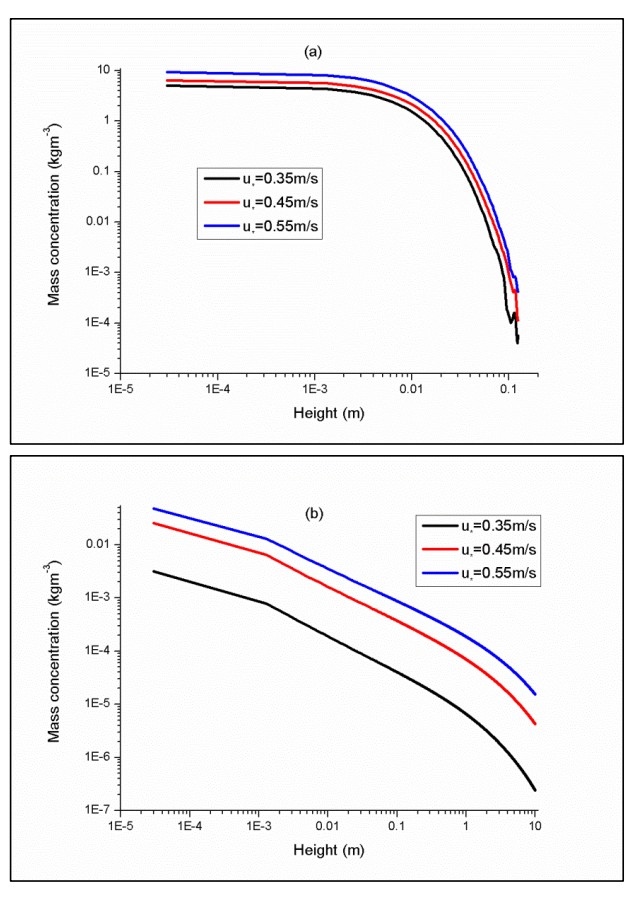

**Figure 10: Vertical profiles of mass concentration for saltation and suspension (a: saltation particles, b: suspended particles)**





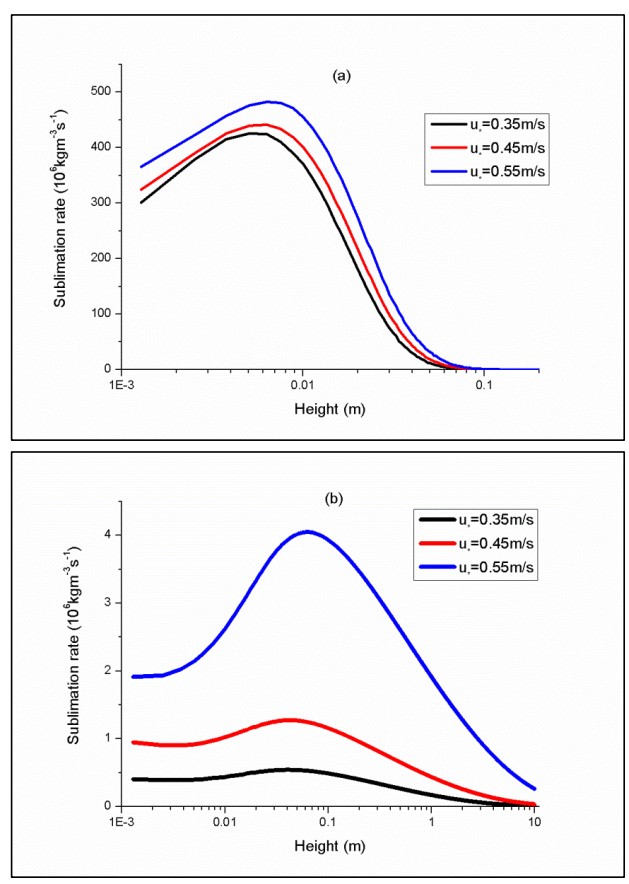

**Figure 11: Vertical profiles of sublimation rate for saltation and suspension (a: saltation particles; b:**

**suspended particles)**

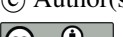


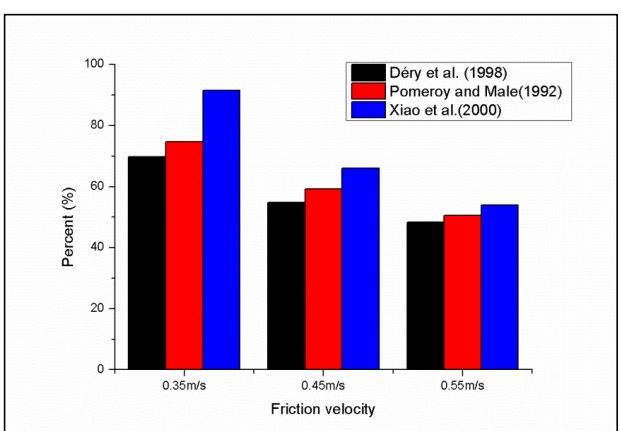

**Figure 12: The ratio of sublimation mass below three heights to the total (the sublimation mass below a**

**height is the sublimation mass that was ignored by other's model , such as Déry et al. (1998), Pomeroy and**

**Male (1992), and Xiao et al. (2000).)**



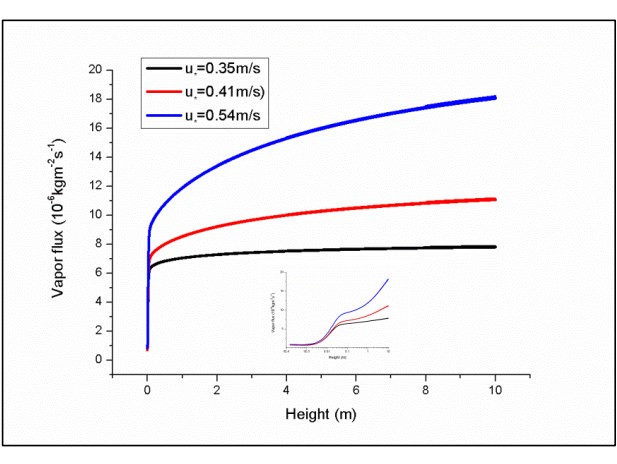

**Figure 13: Vertical profiles of vapor flux**