# Peer review of "The significance of vertical moisture diffusion on"

_The Cryosphere, 2017_

## Referee Comment (RC1) · Anonymous Referee #1 · 5 Aug 2017

The authors present a relatively comprehensive snow drift model, taking into consideration of vertical diffusion of humidity. The results are compared with published data. It is shown that the results are at least qualitatively consistent with the observations and in some aspect also quantitatively consistent. I see considerable value in the further development of the model to a full scale comprehensive model. This model is a very good starting point, as it already has all the ingredients.

The text can be improved. The introduction can be shorter. The very first sentence in the abstract is a very long sentence trying to say too much. Also, the model formulation can be made more concise, e.g., Equation (1). It is unnecessary to write it in such a

complex way.

The discontinuity of the model results is somewhat surprising, like in Figure 2a. The authors should explain what makes the model to behave like that and how it can be improved.

I hope the authors can give a critical assessment of their model and point out the potential for further development.

In general, I find the work very interesting and represents a very useful contribution to snow drift modelling. As it has been revised and many question from the earlier review reports have been considered, I think the paper is now of good quality.
* * *

---

## Referee Comment (RC2) · Anonymous Referee #2 · 7 Aug 2017

This work deals with an important topic, and is one of the few studies that deal with the blowing snow sublimation over near-surface region. The results show that the sublimation will still exist in near-surface region in the fully developed blowing snow, and the mass of sublimation in near-surface region could account for even more than half of the total. The manuscript is laid out in a clear and straightforward manner and adds something new to the physical understanding of the behavior of the snow distribution and transport of snow in the polar, glacier and snowfields etc.. This kind of manuscript is very rare and always of interest, and should be published. However, there are a few points that the authors might consider.

[Figure]

Major points (1) Does the snow sublimation in near-surface region have an impact on the mass and movement of snow particles? That is, Does the change in m also go into equations (4)- (6)? (2) According to the authors, the blowing snow sublimation will reduce the air temperature. It is not clear how the effects of temperature change and the flow field are related in your simulation. (3) Usually the snow particles in the air are divided into suspension and saltation particles, and you seems to distinguish them simply by height. Please explain the reason. (4) Fig. 12 shows that snow sublimation occurs mainly in the near-surface region. It seems contradictory that in Fig. 13 the water vapor flux in the upper air is larger than that in the near-surface region. (5) All the results in Figure 4 don't include the results of saltation particles sublimation, but why the results of this paper is larger than that of xiao et al.. (6) This manuscript refers that there is a negative feedback effect in the blowing snow sublimation. Actually Figure 9 shows that the saltation particles sublimation does have a significant negative feedback effect, but you did not take into consideration of the feedback effect of sublimation of the suspended snow particles? (7) The writing proficiency of this manuscript need to improve because there are some writing errors in this paper. For example, the friction wind speeds in Figure 7 and Figure 8 are not expressed by the same symbol. In the first sentence of the abstract "Drifting snow sublimation is a physical process containing phase change and heat change. . .", the words "of the drifting snow" should be deleted.

---

## Author Comment (AC1) · 8 Aug 2017

Dear reviewer, We'd like to thank you for your insightful comments and positive evaluation of our work. We have studied your comments carefully and will do our best to revise and improve the manuscript. A point to point responds to the reviewer's comments are listed as following: Item 1: The authors present a relatively comprehensive snow drift model, taking into consideration of vertical diffusion of humidity. The results are compared with published data. It is shown that the results are at least qualitatively consistent with the observations and in some aspect also quantitatively consistent. I see considerable value in the further development of the model to a full scale com-

Interactive
comment

prehensive model. This model is a very good starting point, as it already has all the ingredients. Response: Thanks for the positive comments. Our goal is to develop a more comprehensive model considering the sublimation of both saltation and suspension particles in the atmospheric turbulent boundary layer in the future.

Item 2: The introduction can be shorter. The very first sentence in the abstract is a very long sentence trying to say too much. Also, the model formulation can be made more concise, e.g., Equation (1). It is unnecessary to write it in such a complex way. Response: Following the reviewer's suggestion, we will simplify the introduction and model formations in the revised manuscript.

Item 3: The discontinuity of the model results is somewhat surprising, like in Figure 2a. The authors should explain what makes the model to behave like that and how it can be improved. Response: The discontinuity is at a height of about 0.1m in Fig.2a. It can be seen from Fig. 10a that 0.1m is approximately equal to the maximum height of the saltation particles, and snow particles near the height of 0.1m is rare. Therefore the randomness of snow particles' number and their sizes at 0.1m is relatively large, which leads to the discontinuity of snow mass concentration. This problem is more serious in case the wind speed is smaller, for the smaller the wind speed is, the fewer number of snow particles in the air (See Fig.2a). It's much improved when the wind speed is higher (see Fig.2c). We will explain this phenomenon in the revised manuscript.

Item 4: I hope the authors can give a critical assessment of their model and point out the potential for further development. Response: For the future development of the model, we will: (1) extend the model to three dimensions and take into consideration of the effects of turbulence on the sublimation of both saltation and suspension particles in the atmospheric turbulent boundary layer, which will lead to a more accurate and realistic model; (2) propose a parametric model of the blowing snow sublimation, which will provide parameterized values for the mesoscale climate model of the polar ice sheet, the alpine glacier, snowy in the high latitude and so on.
Item 5: In general, I find the work very interesting and represents a very useful contribution to snow drift modelling. As it has been revised and many question from the earlier review reports have been considered, I think the paper is now of good quality. Response: Thank you for your affirmation.

Once again, thank you very much for your comments and suggestions. Best regards Ning Huang and Guanglei Shi

---

## Author Comment (AC2) · 12 Aug 2017

Dear reviewer, Thank you for your comments concerning our manuscript entitled 'The significance of vertical moisture diffusion on drifting snow sublimation near snow surface'. We are grateful to the comments on our manuscript and carefully considered every comment, and will make cautious revision accordingly. Below are our point-to-point responses.

Item 1: Does the snow sublimation in near-surface region have an impact on the mass and movement of snow particles? That is, Does the change in m also go into equations (4)- (6)? Response: Thanks for the comment. We do have calculated the impact of

sublimation on the snow particles, and the results show that the loss in mass of a single particle due to sublimation during its whole movement process is less than 0.1% of its own mass. Thus, the change in m didn't go into equations (4)-(6).

Item 2: According to the authors, the blowing snow sublimation will reduce the air temperature. It is not clear how the effects of temperature change and the flow field are related in your simulation. Response: The temperature drop caused by snow sublimation is generally very small and does not exceed 2K. We verified that the temperature change of 2K has little effect on the wind field and it was ignored in our simulation.

Item 3: Usually the snow particles in the air are divided into suspension and saltation particles, and you seem to distinguish them simply by height. Please explain the reason. Response: In Aeolian study, scientists usually define the particles jumping near surface, as saltation particles, which are mainly composed of large particles; define the particles whose movement distance in the air is long, as suspension particles, which are mainly composed of small particles. For simplicity's sake, a critical height is given. That isïïjŇarticles fly higher than the critical height are regarded as suspension, while particles move below the height are considered as saltation. Furthermore, some scientists believed that the blowing snow sublimation in the near-surface region cloud be ignored, so they assumed that relative humidity below the critical height, which was used to distinguish the saltation and suspended particles is 100%. In this paper, we chose three heights defined by other scientists (see Table 3), and calculated the blowing snow sublimation masses below these heights. The results show that all the sublimation masses below the three heights, account for more than half of the total sublimation mass (see Fig. 12). Because the difference of the critical height defined by different scientists very greatly (see Table 3), which made the simulation results produce a big difference. In this manuscript, we distinguish the saltation and suspending particles (Eq.2) based particles' flowing ability of the wind field. The diffusion equation was applied to describe the motion of suspended particles for small snow particles follow the wind field well. The Lagrangian particle tracing method was used to trace the

motion of every large snow particle saltating in the near-surface region.

Item 4: Fig. 12 shows that snow sublimation occurs mainly in the near-surface region. It seems contradictory that in Fig. 13 the water vapor flux in the upper air is larger than that in the near-surface region. Response: Because snow sublimation occurs mainly in the near surface, the humidity will decrease with the height. The water vapor produced by sublimation will be transferred from the lower humidity area to the higher one, and the amount of water vapor flux is determined by the concentration gradient of water vapor, not by the amount of sublimation. Therefore, it is possible that the water vapor flux in the upper air is larger than that in the near-surface region.

Item 5: All the results in Figure 4 don't include the results of saltation particles sublimation, but why the results of this paper is larger than that of xiao et al.. Response: In the simulation of Xiao et al., they considered that the water vapor in the near-surface region was saturated. That is, the humidity in the near-surface region was assumed to be 100%. In our simulation, the humidity in the near-surface region would not attain to 100% because of the vertical transportation of water vapor. Thus, the calculated humidity of this paper is smaller than that of Xiao et al., and the sublimation result of this paper is larger than that of Xiao et al. accordingly.

Item 6: This manuscript refers that there is a negative feedback effect in the blowing snow sublimation. Actually Figure 9 shows that the saltation particles sublimation does have a significant negative feedback effect, but you did not take into consideration of the feedback effect of sublimation of the suspended snow particles? Response: It can be seen from Fig10a, 11a that the mass concentration as well as sublimation rate of the saltation snow particles is very high, so the saltation snow particles sublimation will strongly affect the temperature and humidity of the surrounding air. Therefore, it has a very strong negative feedback effect. However, it can be seen from Fig10b, 11b that both the mass concentration and sublimation rate of the suspension snow particles are much lower, so the effects of suspension snow particles sublimation on air temperature and humidity are very small. Therefore, its negative feedback effect is negligible.

[Figure]

Item 7: The writing proficiency of this manuscript need to improve because there are some writing errors in this paper. For example, the friction wind speeds in Figure 7 and Figure 8 are not expressed by the same symbol. In the first sentence of the abstract "Drifting snow sublimation is a physical process containing phase change and heat change. . .", the words "of the drifting snow" should be deleted. Response: Following the reviewer's suggestion, we have corrected these writing errors. We also found a native English speaker, who is an English teacher in my university to revise the English of this manuscript so that a clear description on the research will be displayed in the revised version.

Once again, thank you very much for your comments and suggestions. Best regards Ning Huang and Guanglei Shi

---

## Short Comment (SC1) · 21 Aug 2017

Prof. Huang and his team made a very interesting job in analyzing drifting snow sublimation. Their results indicate that blowing snow sublimation is 3-4 orders of magnitude higher than at 10m. It is amazing and very useful, because this phenomenon has not been involved in any land surface model, as I know. I believe this job can fill the gap. One minor opinion: more explanation in figure captions may be better for readers.

---

## Author Comment (AC3) · 23 Aug 2017

Dear Dr. Hongyi Li, Thanks for your positive comments, and we're grateful for your valuable suggestion. The point to point responds are listed as following: Item 1: Prof. Huang and his team made a very interesting job in analyzing drifting snow sublimation. Their results indicate that blowing snow sublimation is 3-4 orders of magnitude higher than at 10m. It is amazing and very useful, because this phenomenon has not been involved in any land surface model, as I know. I believe this job can fill the gap. Response: Thanks. In this paper, we just verify the importance of blowing snow sublimation in near-surface region. In further work, we will propose a parametric model,

which can be applied to the land surface models. Item 2: One minor opinion: more explanation in figure captions may be better for readers. Response: Following the Dr. Li's suggestion, we will add some explanation in figure captions. Once again, thank you very much for your comments and suggestions. Best regards Ning Huang and Guanglei Shi

---

## Referee Comment (RC3) · A. Toure (Referee) · 31 Aug 2017

The significance of vertical moisture diffusion on drifting Snow sublimation near snow surface

By Ning Huang and Guanglei Shi

**1) General comments**

This paper presents a blowing snow model development. The model takes into account the saltation of snow during blowing snow events.

The objectives of the paper are clear. Howver the connections between equations are not always clear to the reader. Also, I understand the challenge of finding ground-based observations to valide simulation of the sublimation. But the limited number of observations used here weakens the conclusion made by the authors. In other words the lack of sufficient validation data makes it impossible to say for certain if this model constitute an improvement over the previous models. The paper can also benefit from a detailed review from a native English speaker.

**2) Specific comments**

Line 7 : instead of "drifting snow sublimation" the authors could clear state "sublimation of blowing snow" or "sublimation of transported snow particles"

Line 10:  by "snow sublimation near surface "  Do the authors mean  the sublimation during the saltaion phase or the turbulent suspension phase of the blowing snow or both?

Line 10 -11: I would say that the statement is not exactly correct. There are a few  models that take this sublimation of blowing snow into account (see for example Liston and Sturm, 1998, Essery et al.,  1999)

Line 15: the sentence is not clear

Line 17: How small?

Line 20:  this sentence need rewording

Line 42: need to cite references : "Many researchers….(references)"

Line 43: is "violently" the appropriate scientific word to use here?

Line 48:  I would state this: "However, some researchers (references) found that humidity near surface not to reach saturation in the drifting snow in the field, ….."

Line 68: "But this model can not describe snow particles suspending in upper air." The sentence is awkward and need rewording.

Line 85: Authors should explain why use Flows equation instead of Blowing snow equation in for example Liston et Sturm, 1998.

Line 94: judging criterion?? Do the authors mean "Threshold"

Line 97: the authors should show the connection betwwen this equation and the previous ones (eqs 1 and 2).

Line 121: Please add reference

Line 171: Which particles, please explain.

Line 185: This conclusion is based on only 4 observations of field is a little bit of a stretch. For example, there is no observations on the figures 2 a, b, c.

Line 189: What are those environmental conditions?

Can the authors explain why the difference

What is the difference between the authors approach and that o schmidt?

Line 195: What are the difference between the 4 models apart from that the all neglect sublimation?

Lines 200: - 205: this section is not clear and need to be reworded

Line 206: "suspended particles versus various friction velocities" IT should be "for various …" instead of "versus, …"

Line 209: "reach stead" what does that mean? do you mean "plateau" = constant value?

Line 214-216: This sentence is incomprehensible

Line 219: Need rewording

Line 221: Need rewording too

Line 227: "reach steady" is not an appropriate phrase to use in my view.

Line 230: need rewording

Line 246: The sencence need rewording

Line 270: I don't think this statement is true. Many models do take the sublimation into account

Line 275: Pomeroy et Male (1992)??? Vionnet et al. (year???)

Line 284: Could the authors cite the study that neglected the saltation?

Figure 13: The smaller figure is not readable

Line 307: "Bellowing snow" what does that need?

---

## Author Comment (AC4) · 4 Sep 2017

Dear Dr. Ally Toure, Thank you for your comments concerning our manuscript entitled 'The significance of vertical moisture diffusion on drifting snow sublimation near snow surface'. We deeply appreciate the time and effort you have spent in reviewing our manuscript. We will carefully consider each comment, and make cautious revision accordingly. Below are our point-to-point responses.

Item 1: The objectives of the paper are clear. However the connections between equations are not always clear to the reader. Response: Thanks for the comment. Following your suggestion, we will add more descriptions on the connections between equations

[Figure]

in the revised manuscript.

Item 2: Also, I understand the challenge of finding ground-based observations to valide simulation of the sublimation. But the limited number of observations used here weakens the conclusion made by the authors. In other words the lack of sufficient validation data makes it impossible to say for certain if this model constitute an improvement over the previous models. Response: Thanks. Just as you said, ground-based observations sublimation observations are few. However, we still found some experimental results that could be used to validate our model. For example, we compared our simulated snow sublimation rate with that of Schmidt's observational results (see Fig. 3). We also compared the simulated snow mass concentration with that of Pomeroy and Male (1992) to indirectly validate our model (see Fig. 2).

Item 3: Line 7 : instead of "drifting snow sublimation" the authors could clear state "sublimation of blowing snow" or "sublimation of transported snow particles". Response: Thanks for your useful suggestion. We will replace the phrase drifting snow sublimation with sublimation of blowing snow in the revised manuscript.

Item 4: Line 10: by "snow sublimation near surface " Do the authors mean the sublimation during the saltaion phase or the turbulent suspension phase of the blowing snow or both? Response: In this manuscript, "snow sublimation near surface" includes the sublimation of both saltation particles and turbulent suspension particles in the region which is close to the snow bed.

Item 5: Line 10 -11: I would say that the statement is not exactly correct. There are a few models that take this sublimation of blowing snow into account (see for example Liston and Sturm, 1998, Essery et al., 1999). Response: Thanks for the comment. You are right. Although in most of models snow sublimation near surface was ignored, some models did consider the sublimation of near bed. But in these models the value of sublimation near surface is only a rough estimate by some empirical formula based on assumptions. We will comment on previous work with more precise sentences in

the revised manuscript.

Item 6: Line 15: the sentence is not clear. Response: Thanks for your suggestion. We will modify the sentence so that its meaning will be clearly expressed in the revised manuscript.

Item 7: Line 17: How small? Response: From Fig. 12, we can see that the mass of snow sublimation near surface accounts for even more than half of the total when the friction wind velocity is less than about 0.55 m/s.

Item 8: Line 20: this sentence need rewording Response: Following your suggestion, we will reword this sentence in the revised manuscript.

Item 9: Line 42: need to cite references : "Many researchers….(references)" Response: Thanks for your carefully reviewing of the manuscript. We will add some relevant references in the revised manuscript.

Item 10: Line 43: is "violently" the appropriate scientific word to use here? Response: Thanks. As your suggestion, the sentences in line 43-45 will be modified in the revised manuscript as: Many researchers believed that the snow particles sublimation near surface would be great at the early stage of drifting snow process. However, the high concentration of snow particles near surface would result in a rapid air temperature decrease and humidity increase. Therefore the humidity would reach saturation quickly near surface, and the sublimation would stop at the saturated layer of humidity.

Item 11: Line 48: I would state this: "However, some researchers (references) found that humidity near surface not to reach saturation in the drifting snow in the field, ….." Response: Thanks. Following your suggestion, we will add some references in the revised manuscript.

Item 12: Line 68: "But this model can not describe snow particles suspending in upper air." The sentence is awkward and need rewording. Response: Thanks. Following your suggestion, we will modify this sentence in the revised manuscript.

Item 13: Line 85: Authors should explain why use Flows equation instead of Blowing snow equation in for example Liston et Sturm, 1998. Response: In order to accurately calculate the sublimation mass of snow particles, we need to know detailed information of each snow particles in the air (including particle size, relative velocity of particles to the wind speed, etc.). These data can't be directly obtained from the blowing snow equation. But they can be calculated by the flow equation and the snow particle motion equation.

Item 14: Line 94: judging criterion?? Do the authors mean "Threshold". Response: "judging criterion" is the criterion for judging whether a particle is a saltation particle or a suspended particle.

Item 15: line 97: the authors should show the connection between this equation and the previous ones (eqs 1and 2). Response: Thanks. This equation is used to calculate the final sedimentation velocity, which is a parameter in Equation 2. We will explain it in the revised manuscript.

Item 16: Line 121: Please add reference. Response: Thanks. As your suggestion, we will add some references in the revised manuscript.

Item 17: Line 171: Which particles, please explain. Response: Thanks. The snow particle size distribution is that we used in the blowing snow model. We will explain it in the revised manuscript.

Item 18: Line 185: This conclusion is based on only 4 observations of field is a little bit of a stretch. For example, there is no observations on the figures 2 a, b, c. Response: The reviewer is right. Because of the limited observational conditions, only a few observations are generally available. And we can only validate our model with such a limited number of observations.

Item 19: Line 189: What are those environmental conditions? Can the authors explain why the difference What is the difference between the authors approach and that o

schmidt? Response: The conditions used in our simulations are the same as those reported by Schmidt. And we will add the environmental conditions in the revised manuscript. Actually the conditions in the field are much complex and changing fast. Therefore it is almost impossible that the results of numerical simulation and field observation results are exactly the same. Nevertheless, it can be seen from Fig. 3 that the two results are relatively consistent, so we think that our model is reliable.

Item 20: Line 195: What are the difference between the 4 models apart from that the all neglect sublimation? Response: The differences among these four models mainly includes structure of the models, numerical methods, meteorological field treatment and the parameterization schemes although they are based on common physical concepts. Detailed information can be found in the paper by Xiao et al. (2000).

Item 21: Lines 200: - 205: this section is not clear and need to be reworded Response: Thanks. As your suggestion, we will reword this sentence.

Item 22: Line 206: "suspended particles versus various friction velocities" IT should be "for various . . ." instead of "versus, . . ." Response: Thanks. As your suggestion, we will modify this word.

Item 23: Line 209: "reach stead" what does that mean? do you mean "plateau" = constant value? Response: Yes, it means that the values of parameters, such as mass of saltation particles and suspension particles will not change with time.

Item 24: Line 214-216: This sentence is incomprehensible Line 219: Need rewording Line 221: Need rewording too Line 227: "reach steady" is not an appropriate phrase to use in my view. Line 230: need rewording Line 246: The sencence need rewording Response: As your suggestion, we will reword these sentences. Actually, we asked a native English speaker, who is an English teacher in my university will revise the English of this manuscript.

Item 25: Line 270: I don't think this statement is true. Many models do take the sublimation into account Response: Thanks for the comment. You are right. Although in most of models snow sublimation near surface was ignored, some models did consider the sublimation of near bed. But in these models the value of sublimation near surface is only a rough estimate by some empirical formula based on assumptions. We will Comment on previous work with more precise sentences in the revised manuscript.

Item 26: Line 275: Pomeroy et Male (1992)??? Vionnet et al. (year???) Response: Thanks for your comment. We will add the year in the revised manuscript.

Item 27: Line 284: Could the authors cite the study that neglected the saltation? Response: Thanks. As your suggestion, we will add the references in the revised manuscript.

Item 28: Figure 13: The smaller figure is not readable Response: Thanks for your comment. We will modify this figure.

Item 29: Line 307: "Bellowing snow" what does that need? Response: Thanks for your comment. We will modify this word.

Once again, thank you very much for your comments and suggestions. Best regards Ning Huang and Guanglei Shi

―――――――――――――――――――

---

## Author Response (AR1)

**A point-by-point response to the reviews**

Dear reviewers,

We sincerely thank you for the efforts you have made in reviewing our manuscript. Your insightful comments and positive evaluation of our work are really appreciated. We have studied your comments carefully and have revised and improved the manuscript accordingly. A point to point responds to the reviewer's comments are listed as following:

**Reponses to RC1**

**Item 1: The authors present a relatively comprehensive snow drift model, taking into consideration of vertical diffusion of humidity. The results are compared with published data. It is shown that the results are at least qualitatively consistent with the observations and in some aspect also quantitatively consistent. I see considerable value in the further development of the model to a full scale comprehensive model. This model is a very good starting point, as it already has all the ingredients.**

**Response:** Thanks for the positive comments. Our goal is to develop a more comprehensive model considering the sublimation of both saltating and suspended particles in the atmospheric turbulent boundary layer in the future, which is depicted in line 389-395 of page in the revised manuscript.

**Item 2: The introduction can be shorter. The very first sentence in the abstract is a very long sentence trying to say too much. Also, the model formulation can be made more concise, e.g., Equation (1). It is unnecessary to write it in such a complex way.**

**Response:** Following the reviewer's suggestion, we have simplified the introduction in line 7-10 of page 1 and model formations in line 104-105 of page 4 in the revised manuscript.

**Item 3: The discontinuity of the model results is somewhat surprising, like in Figure 2a. The authors should explain what makes the model to behave like that and how it can be improved.**

**Response:** The discontinuity is at a height of about 0.1m in Fig.2a. It can be seen from Fig. 10a that 0.1m is approximately equal to the maximum height of the saltating particles, and snow particles near the height of 0.1m is rare. Therefore the randomness of snow particles' number and their sizes at 0.1m is relatively large, which leads to the discontinuity of snow mass concentration. This problem is more serious in case the wind speed is smaller, for the smaller the wind speed is, the fewer number of snow

particles in the air (See Fig.2a). It's much improved when the wind speed is higher (see Fig.2c). We have explained this phenomenon in the revised manuscript in line 235-241 of page 10 in the revised manuscript.

**Item 4: I hope the authors can give a critical assessment of their model and point out the potential for further development.**

**Response:** For the future development of the model, we will: (1) extend the model to three dimensions and take into consideration of the effects of turbulence on the sublimation of both saltating and suspended particles in the atmospheric turbulent boundary layer, which will lead to a more accurate and realistic model; (2) propose a parametric model of the blowing snow sublimation, which will provide parameterized values for the mesoscale climate model of the polar ice sheet, the alpine glacier, snowy with the high latitude and so on. We have added this content in line 389-395 of page 15 in the revised manuscript.

**Item 5: In general, I find the work very interesting and represents a very useful contribution to snow drift modelling. As it has been revised and many question from the earlier review reports have been considered, I think the paper is now of good quality.**

**Response:** Thanks you the positive comments.

**Reponses to RC2**

**Item 1: This work deals with an important topic, and is one of the few studies that deal with the blowing snow sublimation over near-surface region. The results show that the sublimation will still exist in near-surface region in the fully developed blowing snow, and the mass of sublimation in near-surface region could account for even more than half of the total. The manuscript is laid out in a clear and straightforward manner and adds something new to the physical understanding of the behavior of the snow distribution and transport of snow in the polar, glacier and snowfields etc.. This kind of manuscript is very rare and always of interest, and should be published.**

**Response:** Thanks for the positive comments. Our results in this paper show that the sublimation of blowing snow particles can't be ignored. We wish that the blowing snow sublimation near surface can be taken seriously in the future study.

**Item 2: Does the snow sublimation in near-surface region have an impact on the mass and movement of snow particles? That is, Does the change in m also go into equations (4)- (6)?**

**Response:** Thanks for the comment. We do have calculated the impact of sublimation on the snow particles, and the results show that the loss in mass of a single particle due to sublimation during its whole movement process is less than 0.1% of its own mass. Thus, the change in m didn't go into equations (4)-(6).

**Item 3: According to the authors, the blowing snow sublimation will reduce the air temperature. It is not clear how the effects of temperature change and the flow field are related in your simulation.**

**Response:** The temperature drop caused by snow sublimation is generally very small and does not exceed 2K. We verified that the temperature change of 2K has little effect on the wind field and it was ignored in our simulation.

**Item 4: Usually the snow particles in the air are divided into suspension and saltation particles, and you seem to distinguish them simply by height. Please explain the reason.**

**Response:** In Aeolian study, scientists usually define the particles jumping near surface, as saltation particles, which are mainly composed of large particles; define the particles whose movement distance in the air is long, as suspension particles, which are mainly composed of small particles. For simplicity's sake, a critical height is given. That is, particles fly higher than the critical height are regarded as suspension, while particles move below the height are considered as saltation. Furthermore, some scientists believed that the blowing snow sublimation in the near-surface region cloud be ignored, so they assumed that relative humidity below the critical height, which was used to distinguish the saltating and suspended particles. In this paper, we chose three heights defined by other scientists (see Table 3), and calculated the blowing snow sublimation masses below these heights. The results show that all the sublimation masses below the three heights, account for more than half of the total sublimation mass (see Fig. 12).

Because the difference of the critical height defined by different scientists very greatly (see Table 3), which made the simulation results produce a big difference. In this manuscript, we distinguish the

saltation and suspending particles (Eq.2) based particles' flowing ability of the wind field. The diffusion equation was applied to describe the motion of suspended particles for small snow particles follow the wind field well. The Lagrangian particle tracing method was used to trace the motion of every large snow particle saltating in the near-surface region.

**Item 5: Fig. 12 shows that snow sublimation occurs mainly in the near-surface region. It seems contradictory that in Fig. 13 the water vapor flux in the upper air is larger than that in the near-surface region.**

**Response:** Because snow sublimation occurs mainly in the near surface, the humidity will decrease with the height. The water vapor produced by sublimation will be transferred from the higher humidity area to the lower one, and the amount of water vapor flux is determined by the concentration gradient of water vapor, not by the amount of sublimation. Therefore, it is possible that the water vapor flux in the upper air is larger than that in the near-surface region.

**Item 6: All the results in Figure 4 don't include the results of saltation particles sublimation, but why the results of this paper is larger than that of xiao et al..**

**Response:** In the simulation of Xiao et al., they considered that the water vapor in the near-surface region was saturated. That is, the humidity in the near-surface region was assumed to be 100%. In our simulation, the humidity in the near-surface region would not attain to 100% because of the vertical transportation of water vapor. Thus, the calculated humidity of this paper is smaller than that of Xiao et al., and the sublimation result of this paper is larger than that of Xiao et al. accordingly.

**Item 7: This manuscript refers that there is a negative feedback effect in the blowing snow sublimation. Actually Figure 9 shows that the saltation particles sublimation does have a significant negative feedback effect, but you did not take into consideration of the feedback effect of sublimation of the suspended snow particles?**

**Response:** It can be seen from Fig10a, 11a that the mass concentration as well as sublimation rate of the saltating snow particles is very high, so the saltating snow particles sublimation will strongly affect the temperature and humidity of the surrounding air. Therefore, it has a very strong negative feedback effect. However, it can be seen from Fig. 10b, 11b that both the mass concentration and sublimation

rate of the suspended snow particles are much lower, so the effects of suspended snow particles sublimation on air temperature and humidity are very small. Therefore, its negative feedback effect is negligible.

**Item 8: The writing proficiency of this manuscript need to improve because there are some writing errors in this paper. For example, the friction wind speeds in Figure 7 and Figure 8 are not expressed by the same symbol. In the first sentence of the abstract "Drifting snow sublimation is a physical process containing phase change and heat change. . .", the words "of the drifting snow" should be deleted.**

**Response:** Following the reviewer's suggestion, we have corrected these writing errors in Fig. 7 in page 25 and in the revised manuscript in line 7-10 of page 1. A native English speaker, who is an English teacher in my university, has revised the English of this manuscript so that a clear description on the research has been displayed in the revised version. All revised sentences are marked by green.

**Reponses to SC1**

**Item 1: Prof. Huang and his team made a very interesting job in analyzing drifting snow sublimation. Their results indicate that blowing snow sublimation is 3-4 orders of magnitude higher than at 10m. It is amazing and very useful, because this phenomenon has not been involved in any land surface model, as I know. I believe this job can fill the gap.**

**Response:** Thanks. In this paper, we just verify the importance of blowing snow sublimation in near-surface region. In further work, we will propose a parametric model, which can be applied to the land surface models.

**Item 2: One minor opinion: more explanation in figure captions may be better for readers.**

**Response:** Following the Dr. Li's suggestion, we have added some explanation in figure 1, 2, 3, 4 in page19-22 in the revised manuscript.

**Reponses to RC3**

**Item 1: The objectives of the paper are clear. However the connections between equations are not always clear to the reader.**

**Response:** Thanks for the comment. Following your suggestion, we have added the calculation process, where the connections between equations are clearly described in line 202-218 of page 9 in the revised manuscript.

**Item 2: Also, I understand the challenge of finding ground-based observations to valide simulation of the sublimation. But the limited number of observations used here weakens the conclusion made by the authors. In other words the lack of sufficient validation data makes it impossible to say for certain if this model constitute an improvement over the previous models.**

**Response:** Thanks. Just as you said, ground-based sublimation observations are very few. However, we still found some experimental results that could be used to validate our model. For example, we compared our simulated snow sublimation rate with that of Schmidt's observational results (see Fig. 3) in page 21 in the revised manuscript. We also compared the simulated snow mass concentration with that of Pomeroy and Male (1992) to indirectly validate our model (see Fig. 2) in page 20 in the revised manuscript.

**Item 3: Line 7 : instead of "drifting snow sublimation" the authors could clear state "sublimation of blowing snow" or "sublimation of transported snow particles".**

**Response:** Thanks for your useful suggestion. We have replaced the phrase drifting snow sublimation with sublimation of blowing snow in the revised manuscript.

**Item 4: Line 10: by "snow sublimation near surface " Do the authors mean the sublimation during the saltaion phase or the turbulent suspension phase of the blowing snow or both?**

**Response:** In this manuscript, "snow sublimation near surface" includes the sublimation of both saltation particles and turbulent suspension particles in the region which is close to the snow bed.

**Item 5: Line 10 -11: I would say that the statement is not exactly correct. There are a few models that take this sublimation of blowing snow into account (see for example Liston and Sturm, 1998, Essery et al., 1999).**

**Response:** Thanks for the comment. You are right. Although in most of models snow sublimation near surface was ignored, some models did consider the sublimation of near bed. But in these models the value of sublimation near surface is only a rough estimate by some empirical formula based on

assumptions. We have added some comments on previous work with more precise sentences in line 10-12 of page 1 in the revised manuscript.

**Item 6: Line 15: the sentence is not clear.**

**Response:** Thanks for your suggestion. We have modified the sentence in line 18 of page 1.

**Item 7: Line 17: How small?**

**Response:** From Fig. 12, we can see that the mass of snow sublimation near surface accounts for even more than half of the total when the friction wind velocity is less than about 0.55 m/s. We have added the specific value of wind velocity in line 21-22 of page 1 in the revised manuscript.

**Item 8: Line 20: this sentence need rewording**

**Response:** Following your suggestion, we have reworded this sentence in line 25-28 of page 1 in the revised manuscript.

**Item 9: Line 42: need to cite references : "Many researchers….(references)"**

**Response:** Thanks for your carefully reviewing of the manuscript. We have added some relevant references in line 56 of page 2 in the revised manuscript.

**Item 10: Line 43: is "violently" the appropriate scientific word to use here?**

**Response:** Thanks. As your suggestion, the sentences in line 43-45 have been modified in line 56-60 of page 2 in the revised manuscript as: Many researchers (Déry et al., 1998; Bintanja, 2001a; Mann et al., 2000) believed that the sublimation of snow particles near surface would be significant at the early stage of drifting snow process. However, the high concentration of snow particles near surface would result in a rapid air temperature decrease and humidity increase. Therefore, the humidity near surface would quickly reach saturation, leading to sublimation ceasing in the layer with saturated humidity.

**Item 11: Line 48: I would state this: "However, some researchers (references) found that humidity near surface not to reach saturation in the drifting snow in the field, ….."**

**Response:** Thanks. Following your suggestion, we have added some references in line 66-67 of page 2

in the revised manuscript.

**Item 12: Line 68: "But this model can not describe snow particles suspending in upper air." The sentence is awkward and need rewording.**

**Response:** Thanks. Following your suggestion, we have modified this sentence in line 89-90 of page 3 in the revised manuscript.

**Item 13: Line 85: Authors should explain why use Flows equation instead of Blowing snow equation in for example Liston et Sturm, 1998.**

**Response:** In order to accurately calculate the sublimation mass of snow particles, we need to know detailed information of each snow particle in the air (including particle size, relative velocity of particles to the wind speed, etc.). These data can't be directly obtained from the blowing snow equation. But they can be calculated by combine the flow equation and the snow particle motion equation. Therefore the Flows equation is used by many scientists, and we added such a reference in line 104-105 of page 4 in the revised manuscript.

**Item 14: Line 94: judging criterion?? Do the authors mean "Threshold".**

**Response:** "judging criterion" is the criterion for judging whether a particle is a saltating particle or a suspended particle.

**Item 15: line 97: the authors should show the connection between this equation and the previous ones (eqs 1and 2).**

**Response:** Thanks. This equation is used to calculate the final sedimentation velocity of the particles, which is a parameter in Equation 2. We have explained it in line 121-122 of page 5, and further explained the connections of all the equations in the calculation processes in line 202-218 of page 9 in the revised manuscript.

**Item 16: Line 121: Please add reference.**

**Response:** Thanks. As your suggestion, we have added some references in line 149 of page 6 in the revised manuscript.

**Item 17: Line 171: Which particles, please explain.**

**Response:** Thanks. The snow particle size distribution is that we used in the blowing snow model. We have explained it in line 200-201 of page 9 in the revised manuscript.

**Item 18: Line 185: This conclusion is based on only 4 observations of field is a little bit of a stretch. For example, there is no observations on the figures 2 a, b, c.**

**Response:** The reviewer is right. Because of the limited observational conditions, only a few observations are generally available to validate our model.

**Item 19: Line 189: What are those environmental conditions? Can the authors explain why the difference What is the difference between the authors approach and that o schmidt?**

**Response:** The conditions used in our simulations are the same as those reported by Schmidt. And we have added the environmental conditions in figure caption of Fig.3 in page 21 in the revised manuscript. Actually the conditions in the field are much complex and changing fast. Therefore it is almost impossible that the results of numerical simulation and field observation results are exactly the same. Nevertheless, it can be seen from Fig. 3 that the two results are relatively consistent, so we think that our model is reliable.

**Item 20: Line 195: What are the difference between the 4 models apart from that the all neglect sublimation?**

**Response:** The differences among these four models mainly include their structures, numerical methods, meteorological field treatment and the parameterization schemes although they are based on common physical concepts. Detailed information can be found in the paper by Xiao et al. (2000).

**Item 21: Lines 200: - 205: this section is not clear and need to be reworded**

**Response:** Thanks. As your suggestion, we have reworded this sentence in line 257-263 of page 11..

**Item 22: Line 206: "suspended particles versus various friction velocities" IT should be "for various …" instead of "versus, …"**

**Response:** Thanks. As your suggestion, we have modified this word in line 266 of page 11.

**Item 23: Line 209: "reach stead" what does that mean? do you mean "plateau" = constant value?**

**Response:** Yes, it means that the values of parameters, such as mass of saltating particles and suspended particles will not change with time.

**Item 24: Line 214-216: This sentence is incomprehensible**

**Line 219: Need rewording**

**Line 221: Need rewording too**

**Line 227: "reach steady" is not an appropriate phrase to use in my view.**

**Line 230: need rewording**

**Line 246: The sencence need rewording**

**Response:** As your suggestion, we have reworded these sentences in 273-279 of page 11, line 281-286 pages 11-12, line 285-288 of page 12, line 294-296 of page 12, line 298-301 of page 12, line 315-317 of page 13. Actually, we asked a native English speaker, who is an English teacher in my university, have revised the English of this manuscript.

**Item 25: Line 270: I don't think this statement is true. Many models do take the sublimation into account**

**Response:** Thanks for the comment. You are right. Although in most of models snow sublimation near surface was ignored, some models did consider the sublimation of near bed. But in these models the value of sublimation near surface is only a rough estimate by some empirical formula based on assumptions. We have made some comments on previous work with more precise sentences in line 343-344 of page14 in the revised manuscript.

**Item 26: Line 275: Pomeroy et Male (1992)??? Vionnet et al. (year???)**

**Response:** Thanks for your comment. We have added the year in line 351 of page 14 in the revised manuscript.

**Item 27: Line 284: Could the authors cite the study that neglected the saltation?**

**Response:** Thanks. As your suggestion, we have added the references in line 361-362 of page 14 in the revised manuscript.

**Item 28: Figure 13: The smaller figure is not readable**

**Response:** Thanks for your comment. The data of the small figure and the lager one in Fig. 13 are same. The only difference between the small figure and the larger one in Fig.13 is that the small one uses the logarithmic coordinates as x coordinates, and the large one use the linear coordinates as x coordinates. We have deleted it in page 31 in the revised manuscript.

**Item 29: Line 307: "Bellowing snow" what does that need?**

**Response:** Thanks for your comment. We have modified this word in line 387 of page 15.

Once again, thank you very much for your comments and suggestions.

Best regards

Ning Huang and Guanglei Shi

**a list of all relevant changes made in the manuscript**

'Snow' has been rewritten as 'snow'.

The Superscript "1" is deleted.

[revised manuscript text omitted]

'so' has been rewritten as 'So they believed that'.

'blowing sublimation' has been rewritten as 'blowing snow sublimation'.

', taking' has been rewritten as 'by taking'.

the" Drifting snow sublimation" have been written as "Sublimation of blowing snow" in the revised manuscript.

'grassland covered by snow' has been rewritten as 'snow-covered grassland'.

', which can describe the movement of small particles well. But the diffusion equation is difficult to describe the movement of large snow particles which are mainly distributed in the near surface area (Déry et al., 1998; Xiao et al., 2000; Vionnet et al. 2014).'

has been rewritten as

'. Although the equation is good on describing the movement of small particles well, but it is difficult to describe the movement of large snow particles which are mainly distributed in the near surface area (Déry et al., 1998; Xiao et al., 2000; Vionnet et al. 2014).'.

'saltation' has been rewritten as 'saltating'.

'on' has been rewritten as 'with'.

'But this model can not describe snow particles suspending in upper air.'

has been rewritten as

'But this model did not take into consideration of to turbulent suspension of snow particles.'.

'all above' has been rewritten as 'all the above'.

'Therefore, a drifting snow model has firstly been built to describe the movement of snow particles of both saltating near surface and suspending in the higher region. Then, a drifting snow sublimation

model has been built the combination of the drifting snow model, a vertical moisture diffusion equation and a heat balance equation. Then drifting snow sublimation with three wind speeds was calculated. The temporal evolution and vertical profiles of temperature, relative humidity, mass concentration of snow particles, snow sublimation rate were analyzed in details. Meanwhile, the proportions of the sublimation mass of saltation snow grains and saltation layer to the total sublimation mass were also given.'

has been rewritten as

'In this study, a drifting snow model was first established to describe the movement of snow particles of both saltating snow particles near surface and suspended snow particles in the higher region. Then, a sublimation model of blowing snow was built in combination of the drifting snow model, a vertical moisture diffusion equation and a heat balance equation. Next, sublimation of blowing snow at three different wind speeds was calculated and the temporal evolution and vertical profiles of temperature, relative humidity, mass concentration of snow particles and snow sublimation rate were analyzed in details. At last, the proportions of the sublimation mass of snow particles near surface to the total sublimation mass were also given.'.

Line 102 of page 4:

'Method' has been rewritten as 'Methods'

Line 103 of page 4:

'Basic Equations of the Flows' has been rewritten as 'Basic flow equations'.

Line 105 of page 4:

The reference '(Nemoto and Nishimura, 2004)' has been added in.

Line 105-108 of page 4:

'Considering a fully developed steady flow field on an infinite polar ice sheet where the changes of wind field in the lateral and flow direction are negligible, the fully developed horizontal direction flow field equation can be obtained according to the theory of mixing length by Prandtl.' has been deleted.

'saltation' has been written as 'saltating'.

'saltation' has been written as 'saltating'.

'suspension' has been written as 'suspended'.

'saltation' has been written as 'saltating'.

'saltation' has been written as 'saltating'.

'$w_s$ is the final sedimentation velocity of the particles which can be calculated by the following equations (Carrier, 1953):'

has been changed to

'$w_s$ is the final sedimentation velocity of the particles which can be calculated by the following equations (Carrier, 1953):'.

'densities' has been modified to 'density'.

'particle' has been modified to 'particles'.

'saltation' has been written as 'saltating'.

'Saltation particle motion equation is as follows (Huang et al., 2011):'

has been written as

'The motion equation of the saltating particles is as follows (Huang et al., 2011),'

'respectively' has been added in this line.

'respectively' has been added in this line.

'relative velocity of movement' has been rewritten as 'movement relative velocity'.

'and' has been rewritten as 'in'.

'respectively' has been added in this line.

'respectively' has been added in this line.

'Basic Equations of Suspended particles' has been written as 'Basic equations of suspended particles'.

'suspension' has been written as 'suspended'.

Line 149 of page 6:

The reference 'Déry and Yau, 1999' has been added in.

Line 152 of page 6:

'grain.' has been rewritten as 'particles, and'

Line 154 of page 6:

' $w'$ is the turbulent fluid velocity in the vertical'

has been rewritten as

' $w'$ is the vertical turbulent fluid velocity'.

Line 155 of page 6:

The word 'and' has been added in.

Line 156 of page 7:

'Aerodynamic Entrainment' has been rewritten as 'Aerodynamic entrainment'.

Line 159 of page 7:

'causing by' has been rewritten as 'due to'.

Line 165 of page 7:

'relative humidity of air' has been rewritten as 'relative air humidity'.

Line 166 of page 7:

'thermal conductivity of air' has been rewritten as 'air thermal conductivity'.

Line 169 of page 7:

'respectively' has been added in this line.

'equation' has been rewritten as 'equations' .

'The heat and humidity equations of air' has been rewritten as 'The air heat and humidity equations'.

'respectively' has been added in this line.

'and' has been added in this line.

'Where' has been rewritten as 'where'.

',' has been rewritten as 'and'.

'saltation' has been written as 'saltating'.

'and' has been added in this line.

'saltation' has been written as 'saltating'.

'saltation' has been written as 'saltating'.

'saltation' has been written as 'saltating'.

[revised manuscript text omitted]

117     by diffusion equation.

**2.2.1 Judging criteria of  saltating and suspended particles**

    The judging criterion of  saltating and suspended particles is as follows (Scott, 1995):

$$\begin{cases} w_s/(ku_*)>1, & \textit{saltation particle} \\ w_s/(ku_*) \le 1, & \textit{suspension particle} \end{cases} \tag{2}$$

where $u_*$ is the friction velocity and $w_s$ is the final sedimentation velocity of the particles which can be calculated by the following equations (Carrier, 1953):

$$w_s = -\frac{A}{D} + \sqrt{\left(\frac{A}{D}\right)^2 + BD}$$
$$A = 6.203\upsilon_a \tag{3}$$
$$B = \frac{5.516\,\rho_p}{8\,\rho_a}g$$

where D is diameter of snow particle, $\upsilon_a$ is air viscosity coefficient, $\rho_p$ is the  density of snow particles, $g$ is the acceleration of gravity.

**2.2.2 Basic equations of  saltating particles**

      motion equation of the saltating particles is as follows (Huang et al., 2011):

$$m\frac{dU_p}{dt} = F_D\left(\frac{U_a - U_p}{V_r}\right) \tag{4}$$

$$m\frac{dV_p}{dt} = -G + F_B + F_D\left(\frac{V_a - V_p}{V_r}\right) \tag{5}$$

$$\frac{dx_p}{dt} = U_p \tag{6}$$

$$\frac{dy_p}{dt} = V_p \tag{7}$$

where $m$ is the mass of snow particle, $G$ is the gravity of snow particle, $U_a$ and $V_a$ are the horizontal and vertical velocity of air, respectively, $U_p$ and $V_p$ are the horizontal and vertical

velocities of snow particle, respectively, $V_r = \sqrt{(U_p - U_a)^2 + (V_p - V_a)^2}$ is the movement relative

velocity of movement of the snow particles and in the flow field, $F_B$ and $F_D$ are the buoyancy and

traction forces of snow particles, respectively, $x_p$ and $y_p$ are the horizontal and vertical positions

of snow particles.

The splash function fitted by Sugiura and Maeno (2000) according to the observations of the low

temperature wind tunnel experiment was chosen,

$$S_v(e_v) = \frac{1}{b^a G(a)} e_v^{a-1} \exp\left(-\frac{e_v}{b}\right) \tag{8}$$

$$S_h(e_h) = \frac{1}{\sqrt{2\pi\sigma^2}} \exp\left[-\frac{(e_h - \mu)^2}{2\sigma^2}\right] \tag{9}$$

$$S_e(n_e) = {}_mC_{n_e} p^{n_e} (1-p)^{m-n_e} \tag{10}$$

where $S_v(e_v)$, $S_h(e_h)$ and $S_e(n_e)$ are the probability distribution functions of the vertical

restitution coefficient $e_v$, horizontal restitution coefficient $e_h$, and the number of grains ejected $n_e$,

respectively.

**2.2.3 Basic  equations of  suspended particles**

The movement of  suspended particles is described by the following vertical diffusion

equation according to horizontal uniformity condition (Déry and Yau, 1999),

$$\frac{\partial q}{\partial t} = \frac{\partial}{\partial y}\left(K_s \frac{\partial q}{\partial y} + w_s q\right) + S \tag{11}$$

where q is the snow particle mass concentration, $K_s$ is the vertical diffusion coefficient, S is the

volume sublimation rate of snow particles, and $K_s = \delta\kappa u_* z$, $\delta$ is as follows (Csanady, 1963),

$$\delta = \frac{1}{\sqrt{1 + \frac{\beta^2 f^2}{w_a^2}}} \tag{12}$$

where $\beta$ is the proportionality constant, $w'$ is the vertical turbulent fluid velocity in the vertical,

and we set $\beta = 1$, and $\overline{w'^2} = u_*^2$.

**2.2.4 Aerodynamic entrainment**

The aerodynamic entrainment equation of Shao and Li (1999) is chosen,

$$N_a = V u_* \left( 1 - \frac{u_{*t}^2}{u_*^2} \right) D^{-3} \tag{13}$$

where $N_a$ is the number of snow particles taking off due to aerodynamic entrainment, $\varsigma$ is a non-dimensional coefficient, approximately equal to $1 \times 10^{-3}$, $u_*$ is the friction velocity, and $u_{*t}$ is the threshold friction velocity.

**2.3 Sublimation formula**

The sublimation formula is as follows (Thorpe and Mason, 1966),

$$\frac{dm}{dt} = \frac{\pi D (RH - 1)}{\dfrac{L_s}{K Nu T_a} \left( \dfrac{L_s}{R_v T_a} - 1 \right) + \dfrac{
[revised manuscript text omitted]

[Figure]

**Figure 13: Vertical profiles of vapor flux**

---

## Author Response (AR2)

Dear editor,

Thank you very much for taking a lot of time to review our manuscript. We have carefully evaluated all the critical comments and thoughtful suggestions, and revised the manuscript accordingly. Below are the point to point responds.

**Item 1: I would like to thank the authors for making substantial efforts to improve the scientific quality of this re-submission. As noted by the reviewers, the paper offers a clear contribution to the literature and the model description is concise and thorough, which should promote replication of this study and the future application of the model by others.**

**Response:** Thanks for the positive comments.

**Item 2: carry out one final edit to improve the english grammar. As noted by one reviewer, a few examples include:**

**- on line 69: "exiting" should be "existing",**

**- suggest rewording terms "sublimation of blowing snow near surface" to be "sublimation of near-surface blowing snow"**

**Response:** Thanks. Following this suggestion, we have reworded the 'exiting' to be 'existing' in line 71 of page 3, and reworded 'sublimation of blowing snow near surface' to 'sublimation of near-surface blowing snow' in line 9, 10-11, 13, 15-16, 19-20 of page 1 in the revised manuscript. We also modified some other errors in line 31 of page 1, line 54 of page 2, line 176-177 of page 8, line 370 of page 14, line 380, 392, 409-410 of page 15, and the friction wind speeds in figure 13 of page 29 in the revised manuscript.

**Item 3: Previous referees have commented on this, but the issue remains. There appears to be a numerical stability issue in the model that appears between 2 sec and 10 sec in the calculation of saltating and suspended particles (i.e., as inferred from the noisy / wavy lines in Fig. 5). That error impacts your estimates of sublimation (e.g., Fig. 9). While I don't think this negatively influences your results, it suggests that there may remain a bug in the model. What is happening in the estimate of entrainment, in that specific time frame, that is causing the noise? If you run the model, say, 1000 times to reproduce Fig 5, and plot the average values of those runs, does the noise still remain? If not, this could point to a numerical solver issue.**

**Response:** The reviewers are right. In our model, the movement of the saltating particles is described by the Lagrangian particle tracing method. The mass change of saltating snow particles in the air is controlled by two processes: one is the movement of snow particles fall into the snow bed, the other one is that of snow particles take off from snow bed. The mass of snow in the air reach stability means these two processes reach dynamic balance. That is, the number of falling particles is roughly equal to that of lifted particles from surface. But it is still possible that there are fluctuations for mass of saltating particles as shown in Fig. 5 and Fig. 9 because of the randomness of particles movement. This phenomenon also occurred in other models using Lagrangian particle tracing method (for example, McEwan I K. Willetts B B. Numerical model of the saltation cloud. Acta Mech. (Suppl.), 1(1991): 53-66; Nemoto, M., and Nishimura, K.: Numerical simulation of snow saltation and suspension in a turbulent boundary layer, J. Geophys. Res., 109, 1933-1943, 2004). However, just as the reviewers said, the curve of average values will be smooth if the model runs many times. We have added some explanations for this phenomenon in line 233-236 of page 10 in the revised manuscript.

**Item 4: Figure 2: I am surprised by the very close agreement between the authors' model estimates of mass concentration and the Pomeroy and Male datum at the lowest height in each panel (i.e., the authors' model exactly overlaps the left-most red dot in each graph). At all other heights, there is some degree of difference, and that difference varies amongst the three panels. Perhaps I do not understand a key model boundary condition. What is the reason for this near-perfect agreement at the lowest measurement height?**

**Response:** In this manuscript, the movements of saltating particles at the lower height are described by Lagrangian particle tracing method. This method traces the motion of every saltating particle and therefore its results are very close to the actual motions of the snow particles. However, for suspended particles at higher height, their motion is controlled by diffusion equations in this manuscript. Due to some assumptions used in the equations, the accuracy of the results might be lower. Even so, the error between the simulation results and experimental results is within an acceptable range.

Once again, thank you very much for your comments and suggestions.

Best regards

Ning Huang and Guanglei Shi

**A list of all relevant changes made in the manuscript**

Line 9, 10-11, 13, 15-16, 19-20 of page 1

'sublimation of blowing snow near surface' has been reworded to be 'sublimation of near-surface blowing snow'.

Line 31 of page 1

'the fluxes of sublimated blowing snow sublimation fluxes'

has been rewritten as

'the fluxes of sublimated snow'.

Line 54 of page 2

'than' has been deleted.

Line 71 of page 3

'exiting' has been reworded to be 'existing'.

Line 176-177 of page 8

'with random particle size and vertical velocity $\sqrt{2GD}$ .'

has been rewritten as

'with a random particle size D and a vertical velocity of $\sqrt{2GD}$ .'.

Line 233-236 of page 10

'It can be seen that there are some fluctuations at 2 sec - 10 sec. This is due to the randomness of particle movement. And it also occurred in other models using Lagrangian particle tracing method (McEwan and Willetts, 1991; Nemoto and Nishinura, 2004).' has been added in the manuscript.

Line 370 of page 14

'.' has been added in the manuscript.

Line 376-377 of page 14-15

A new reference of 'McEwan I K. Willetts B B. Numerical model of the saltation cloud, Acta Mech.(Suppl.), 1, 53-66, 1991.' has been added in the manuscript.

Line 380 of page 15

'.' has been added in the manuscript.

Line 392 of page 15

'L ayer' has been rewritten as 'Layer'.

Line 409-410 of page 15

'Xiao J, Bintanja R, Déry S J, et al. An Intercomparison Among Four Models Of Blowing Snow[J]. Boundary-Layer Meteorology, 2000, 97(1):109-135.' is same as the following reference, and it has been deleted.

Figure 13 in page 29

The friction wind speeds in Figure 13 is wrong, and it should be same as Figure 11.We have modified it in Figure 13.

[revised manuscript text omitted]